# Does audio-visual information result in improved health-related decision-making compared with audio-only or visual-only information? Protocol for a systematic review and meta-analysis

Jemaine E Stacey ![ORCID],[1] Christopher Atkin,[1] Helen Henshaw ![ORCID],[2,3]
Katherine L Roberts,[1] Harriet A Allen,[4] Lucy V Justice ![ORCID],[1] Stephen P Badham ![ORCID] [1]

[1]NTU Psychology, Nottingham Trent University, Nottingham, UK
[2]Hearing Sciences, Mental Health and Clinical Neurosciences, School of Medicine, University of Nottingham, Nottingham, UK
[3]National Institute for Health and Care Research (NIHR), Nottingham Biomedical Research Centre, Nottingham, UK
[4]School of Psychology, University of Nottingham, Nottingham, UK

**Correspondence to**
Dr Jemaine E Stacey;
jemaine.stacey@ntu.ac.uk

## ABSTRACT

**Introduction** Making health-related decisions can be difficult due to the amount and complexity of information available. Audio-visual information may improve memory for health information but whether audio-visual information can enhance health-related decisions has not been explored using quantitative methods. The objective of this systematic review is to understand how effective audio-visual information is for informing health-related decision-making compared with audio-only or visual-only information.

**Methods and analysis** Randomised controlled trials (RCTs) will be included if they include audio-visual and either audio-only or visual-only information provision and decision-making in a health setting. Studies will be excluded if they are not reported in English. Twelve databases will be searched including: Ovid MEDLINE, PubMed and PsychINFO. The Cochrane Risk of Bias tool (V.7) will be used to assess risk of bias in included RCTs. Results will be synthesised primarily using a meta-analysis; where quantitative data are not reported, a narrative synthesis will be used.

**Ethics and dissemination** No ethical issues are foreseen. Data will be disseminated via academic publication and conference presentations. Findings may also be published in scientific newsletters and magazines. This review is funded by the Economic and Social Research Council.
**PROSPERO registration number** CRD42021255725.

## RATIONALE

Individuals often have to make important decisions which can affect their health such as deciding when to seek medical advice or which treatment to pursue. Yates and Patalano define a decision as 'the selection of an action with the aim of producing satisfying outcomes'.[1] To elaborate on this, we define the decision-making process as involving the reviewing and weighting of information, then making a choice between two and more options to reach a conclusion. A related measure of interest is decisional conflict.

## Strengths and limitations of this study

▶ This systematic review is the first to investigate the use of audio-visual information to improve health-related decisions using meta-analysis to synthesise results.
▶ The level of evidence for outcomes will be assessed using the Grading of Recommendations Assessment, Development and Evaluation.
▶ This protocol was developed using the Preferred Reporting Items for Systematic Review and Meta-Analysis Protocols checklist.
▶ Grey literature will be included.
▶ Studies will only be included if they are published in English.

Decisional conflict is defined by LeBlanc *et al* as 'uncertainty about which course of action to take when choice among competing options involves risk, regret, or challenge to personal life values'.[2] We define audio-visual information as the concurrent presentation of auditory and visual information. This includes speech (in person or video recorded) as this is how health information would normally be delivered, for example, during a face-to-face consultation with a doctor.

Two reviews have been conducted in the field of health information provision focusing on audio-visual information.[3 4] Van der Meulen *et al*[3] focused on recall of health information while the most common outcome variables reported in the Wofford *et al* review included knowledge retention and health attitudes.[4] No review has systematically incorporated audio-visual information and decision-making from a health perspective.

Van der Meulen *et al*[3] reviewed 10 studies which included recall of different methods of delivering health information to patients with

cancer. Interventions which used auditory-only information (audio tape of consultation) and visual-only information (typed letter) were included. Overall, the research showed that providing a letter alone or an audio tape alone did not enhance recall of information, compared with a control group who received no additional information. When the letter and audio tape were directly compared, there was no improvement in recall. However, when patients (mean age of 62 years) were given both a letter and audio tape, recall improved compared with the letter only (audio tape only was not included as a comparator). This suggests that the presence of both audio and visual information may improve memory for health information.

In their narrative review, Wofford *et al* identified 26 randomised controlled trials (RCTs) in the literature that used multimedia information presented via a computer for patient education. The authors noted that the majority of interventions involved audio information and only three studies used video.[4] Only two studies used combined audio-visual information (multimedia computer program and printed booklet) and measured decisional conflict. Decisional conflict was measured in three domains: (1) uncertainty regarding choosing between multiple options to make a decision, (2) factors which contribute to uncertainty and (3) perceived effectiveness of the decision-making.[5 6] Overall, audio-visual information improved decisional conflict scores in all three domains.

In particular, it is important to understand how older adults use information to inform health decisions as they will likely have to make important health decisions in later life, at a time when cognitive ageing is impacting decision-making processes.[7 8] Making informed health decisions may also be harder for older adults who live alone, for example, those who have lost a partner, as they may have to make decisions independently.[9] Furthermore, English and Carstensen[10] demonstrated that poor health reduced the age-related positivity bias (where older adults prioritise positive information over negative information to a greater extent than young adults), during health-related decision-making. This indicated that older adults may find health-related decisions particularly difficult at times when those decisions are most crucial. Recent evidence suggests audio-visual information, compared with auditory-only or visual-only information, may disproportionately benefit cognition for older adults relative to young adults.[11 12] Therefore, this review will aim to evaluate if audio-visual information may be an effective approach to improving health outcomes for older adults.

As the decision-making literature is multidisciplinary, we are interested in synthesising the available evidence using meta-analysis in order to better understand how audio-visual information might improve health-related decision-making. We hypothesise that audio-visual information will benefit all patients compared with audio-only or visual-only information, and audio-visual information may be particularly useful for older adults who might have auditory or visual deficits and/or an increased number of health-related decisions to make.

## OBJECTIVES

The current review aims to establish whether providing health information in dual modality (audio-visual) can improve the quality of health-related decisions, compared with single-modality (audio-only or visual-only) information. The objective is: to understand how effective audio-visual information is for informing health-related decision-making compared with audio-only or visual-only information.

## METHODS AND ANALYSIS

This protocol was prepared in accordance with the Preferred Reporting Items for Systematic Review and Meta-Analysis Protocols checklist.

### Eligibility criteria

Peer-reviewed articles will be included, as well as ongoing research where datasets are available (eg, via clinicaltrials.gov, opengrey.eu or the Open Science Framework). The Participant/Intervention/Comparator/Outcome framework will be used to identify papers. Only articles published in English will be considered given time constraints and lack of funds for translation. There will be no date restrictions placed on the literature searches. Study designs will include RCTs.

Studies will be screened for inclusion of audio-visual stimuli, and audio-only and/or visual-only stimuli. Audio-only stimuli can include spoken speech, for example, telephone consultations or audio recordings. Visual-only stimuli can include pictures, animations and written text. The audio-visual stimuli can include videos, television, or a combination of the audio-only and visual-only stimuli listed. The study can include audio-visual information only or a comparison between audio-visual information and information presented in one of the formats listed above. For decision-making, studies could include three distinct stages: a pre-decision phase where information is given to participants, the decision-making stage where the information is considered and weighted, or a problem-solving task is completed. Finally, a clear choice must be made between two and more options. Decision-making must be measured by a minimum of one (or more) of the following: decisional conflict, confidence in decision or quality of decision. Decisions must be made by an individual for themselves not on behalf of others.

### Participants

Research which includes adults aged 18 years and above will be included. If other age groups are included within studies, but data can be separated, those studies will be eligible for inclusion.

### Intervention/interest

Audio-visual health information.

**Table 1** Search strategies for different databases

| # | Ovid: MEDLINE, EMBASE, PsychINFO<br>exp=explode MeSH, .mp=search titles, abstracts & keywords |
|---|---|
| 1 | exp Decision Making/ |
| 2 | exp Educational Programs/ |
| 3 | health*.mp. |
| 4 | audio-visual*.mp. |
| 5 | Multimedia*.mp. |
| 6 | 2 or 4 or 5 |
| 7 | 1 and 3 and 6 |

## Comparator

Audio-only/visual-only health information.

## Outcomes

Health-related decision-making must be measured by one or all of the following: decisional conflict, confidence in decision or quality of decision.

Subgroup analyses: if data allow, we will conduct separate meta-analyses for younger versus older adults or include age group as a subgroup analysis.

## Information sources

Databases to be searched: Cochrane Database of Systematic Reviews; Database of Abstracts of Reviews of Effects; Cochrane Central Register of Controlled Trials; Cochrane database of methodology reviews; Cochrane methodology register: EBSCO, Ovid EMBASE, Ovid MEDLINE, PubMed, PsychINFO, Scopus and Web of Science (Science and Social Science Citation Index).

Articles identified from the searches will be managed in Refworks to deduplicate citations. Abstracts will then be uploaded to Covidence for screening. Preliminary searches to test search terms were carried out in October 2021; the full search will be conducted in March 2022, and the searches will be repeated before publication, with the estimated submission date in December 2022.

Table 1 shows the search strategies developed with input from a specialist librarian.

## Article selection process

The review team will be JES, CA and HH. Two reviewers (JES and CA) will independently screen the titles and abstracts of retrieved studies against the inclusion/exclusion criteria. If there is any disagreement between the two reviewers, a third reviewer will be involved to reach a decision. If there is not enough information provided in the titles and abstracts to know if it should be included, the full texts will be screened. In the next step of article selection, the full texts will be screened by both reviewers, and a third reviewer will be involved if there is any disagreement. The reference lists of included articles will be checked for relevant research, and the citations of included research articles will be entered into Google Scholar to check for any new or relevant data. We have included the Cochrane Database of Systematic Reviews so that the reference lists of review articles can be checked to ensure relevant articles are captured.

## Data extraction process

JES will be responsible for the creation of the data extraction form. The data extraction process will be subject to piloting by both reviewers. The data from each study will be extracted separately by JES and CA and then compared. A third reviewer will be involved if there is any disagreement.

## Data items

The data extracted will include the aim, comparator, study design (eg, RCT), the decision-making task (eg, choosing a treatment option), setting, conflicts of interest, demographic information about the population, randomisation process, effect sizes reported for the primary outcomes: decision-making measures (eg, decisional conflict/effectiveness/confidence/N of people who made a decision), and secondary outcome: knowledge of health information, any missing outcome data and the selection of the reported result. The authors will be contacted via email if sufficient detail is not reported; and if a response is not received, this will be noted. If data are only reported via figures, then WebPlotDigitizer (http:// arohatgi.info/ WebPlotDigitizer/ app/) will be used to extract the data from figures. A third reviewer will be involved if there is any disagreement between reviewers regarding data extracted from plots.

## Study risk of bias assessment

The reviewers JES and CA will assess risk of bias at the study level for each RCT identified using the Cochrane Risk of Bias tool (V.7) which allows a rating to be assigned of either low risk, unclear or high risk of bias (https:// methods.cochrane.org/risk-bias-2).

## Effect measures

We will express the size of the difference in decision-making for audio-visual health information and a control (audio-only or visual-only) comparison in terms of the standardised mean difference (SMD; Cohen's *d*). We will use the SMD as we expect that decision-making will be measured using different outcome measures. The calculation of the effect size will use the pooled SD (between groups) or the SD of the differences (within group). We will also report the 95% CI for each SMD. SMDs will be corrected using Hedges' *g* which corrects for bias caused by small sample sizes.[13] An effect size greater than 0 indicates that a larger effect was observed for the intervention group compared with the control group. A positive effect size would represent an improvement in health decisions, for example, a reduction in decisional conflict or increased confidence in decision-making.

## Synthesis methods

For eligible studies, if effect sizes are not reported, they will be calculated from means and SDs, or from analysis of

variance or t-tests. Where sufficient data can be extracted, we will conduct random-effects meta-analyses of the SMDs, as we anticipate significant heterogeneity across interventions between studies. The studies will be weighted using the inverse variance method.

Heterogeneity across studies will be examined using the $I^2$ statistic and significance tested using a $\chi^2$ test. If heterogeneity is found, a prediction interval for the true intervention effects will be calculated.[14] We will also check for outliers and conduct an influencer analysis to see if removing any outliers/influential studies affects the $I^2$. If data allow, we will include a subgroup analysis of age group (eg, young adults 18–30 years and older adults 65–80 years). We will include risk of bias as a moderator in the analysis. Analysis will be conducted in R using the Metafor package.[15 16]

A narrative synthesis will be used for studies which cannot be meta-analysed including the study design, sample size, type of multimedia information used and findings (improvements for single modality and dual-modality health information).

JES will keep an ongoing record of any changes to the protocol via PROSPERO, and any deviations from the protocol will be included in the final published version of the systematic review.

### Reporting bias assessment

Funnel plots will be used to assess reporting bias and the Egger *et al*[17] test will be used to assess funnel plot asymmetry.

### Certainty assessment

The level of evidence for results will be assessed for each outcome using the Grading of Recommendations Assessment, Development and Evaluation approach which allows evidence to be rated as either low, moderate or high.

### ETHICS AND DISSEMINATION

This systematic review and meta-analysis does not raise any ethical issues. Results will be disseminated via scientific peer-reviewed journal articles, scientific magazines and conference presentations.

**Acknowledgements** Thanks to Sharon Potter for guidance on the systematic review process and help developing the search strategy.

**Contributors** JES, HH, KLR, HAA, CA and SPB all contributed to the idea for the systematic review. JES and HH led the development of the systematic review protocol. JES, HH and CA drafted the eligibility criteria and study selection plans. LVJ aided in the development of the meta-analysis plans. HH, KLR, HAA and SPB provided feedback on the manuscript and all authors approved the final draft of the protocol. JES is the guarantor of the review.

**Funding** This work was funded by the Economic and Social Research Council (ESRC) (grant number ES/V000071/1) and Open Access funding was provided by NTU Psychology, Nottingham Trent University, Nottingham, UK.

**Competing interests** None declared.

**Patient and public involvement** Patients and/or the public were not involved in the design, or conduct, or reporting, or dissemination plans of this research.

**Patient consent for publication** Not required.

**Provenance and peer review** Not commissioned; externally peer reviewed.

**ORCID iDs**
Jemaine E Stacey http://orcid.org/0000-0003-4035-712X
Helen Henshaw http://orcid.org/0000-0002-0547-4403
Lucy V Justice http://orcid.org/0000-0003-3394-2283
Stephen P Badham http://orcid.org/0000-0002-6890-102X

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
