## [Reviewer comments · BMJ Open]

ARTICLE DETAILS

TITLE (PROVISIONAL)	Does audio-visual information result in improved health-related decision making compared to audio-only or visual-only information? Protocol for a systematic review and meta-analysis
AUTHORS	Stacey, Jemaine; Atkin, Christopher; Henshaw, Helen; Roberts, Katherine L.; Allen, Harriet; Justice, Lucy; Badham, Stephen

VERSION 1 – REVIEW

REVIEWER	Knox, Liam The University of Sheffield, Department of Neuroscience
REVIEW RETURNED	29-Nov-2021

GENERAL COMMENTS	This research proposal is well-written and justified, with the methods exceptionally well documented and appropriate to meet the objectives. The authors should be congratulated on the transparency of this protocol. I have a total of five suggestions for the authors, four of which only relate to providing greater clarity, and one methodological query. All five are relatively minor. Rationale 1. Page 3, line 30 and 33: In the description of Wofford, et al., you describe that the main outcome was decisional conflict. However, you do not provide a definition to aid readers. This could be added in parentheses after the first mention. Much in the way you define positivity bias on page 3, line 40. 2. Page 3, paragraph beginning on line 49: Most of this information appears to be a repeat of what is included in the paragraph beginning line 35. I would suggest that you either combine the two paragraphs or simply remove the second, as you already describe the importance and rationale behind your systematic review. 3. Hypothesis: your hypothesis is well based upon the literature that you describe; however, you should include the comparator within this. For example, "...audio-visual information would be more beneficial for all patients in comparison to X". If you wanted to go further, you could follow the PICO (participants, intervention, comparison, outcome) format, although this may not be necessary for describing a systematic review hypothesis. Methods and analysis 4. Page 5, line 35: You state that you will extract information on the randomisation process, where this would appear to indicate that you may only be including randomised controlled trials. Whether this is correct or not, a sentence could be added to your 'eligibility criteria'
---

	section to provide clarity in this regard. 5. Effect measures: Is it likely that your included studies will have identical group sample sizes? If not, it would be better to calculate your effect sizes by adjusting your standard deviation with weights for the sample size, or using a correction for the positive bias of Cohen's d (as opposed to simply pooling SD). For the purposes of transparency, it would also be beneficial to explicitly state what effect size you are calculating (Cohen's d, Hedges' g, or Glass' delta). I wish you luck with your review.
--	--

REVIEWER	Aquino, Maria Raisa Newcastle University
REVIEW RETURNED	10-Dec-2021

GENERAL COMMENTS	Thank you for the opportunity to review this protocol for a systematic review investigating the effectiveness of audio-visual information on health-related decision-making. It is a well-written and designed protocol for a systematic review with meta-analysis. There are a few aspects of this protocol which need clarification and further detail, outlined below. I hope this will be useful to you. Abstract: -No need to list all databases you will be searching in the abstract. Easier to read if you state now many databases you will search with a few examples. Introduction: -The rationale for this review is clearly set out. Methods and analysis: -You mentioned including ongoing research where datasets are available (page 4, line 16), but have stated in your article summary (page 2, line 52) that you would exclude grey literature. Please can you clarify this? -Regarding information sources, you included the Cochrane Database for Systematic Reviews. Will you be including systematic reviews? If so this is not clear and will have implications on your analysis plan. You are also planning on including the NHS Economic Evaluation Database, and the relevance of this database to your review aim is unclear. Will you be including a synthesis of the cost-effectiveness of audio-visual information in health-related decision making? If so please include plans for this. Finally I wondered if you would do citation searching, or if not, perhaps comment on why you feel this is an unnecessary step. -You might wish to consider what is the added value of searching both MEDLINE and PubMed, or whether the MEDLINE search plus all the other databases (n= 11) you will be searching will be sufficient to capture the extant literature on this topic. The majority of PubMed contain MEDLINE records/citations. -The search strategy is very brief and could benefit from the input of an information specialist/specialist librarian. Although this is presented as the search strategy for OVID, this will need tweaking for EMBASE as well as MEDLINE. For example: 'exp Decision Making/' returns null results when searching OVID MEDLINE. I think
---

	the search strategy presented is also missing a lot of terms related to decision-making, e.g. decision support tools, decision aids, decision support techniques, choice behaviour etc. I strongly recommend that the authors revisit their search strategy in collaboration with an information specialist. This will help to ensure that your review captures as much of the relevant literature. -The article selection process needs further detail. At the moment the full-text screening stage is not described. You have indicated that where there is limited information in the abstract that you would access full-texts. However full-text screening is a separate stage to title and abstract screening. Full-texts of all studies included at title and abstract screening need to be screened to ensure these meet your full eligibility criteria. -Data items look good, however I wondered if it would be useful to include what are the decision-making tasks in these studies? -On the meta-analysis, it would be useful for the authors to comment on how heterogeneity will be addressed. General feedback on manuscript: -Full stops in citations with et al. missing, e.g., page 3 line 11: Van der Meulen et al [2]. -Page 3, lines 49-53 are a repetition of page 3, lines 35-39. Please check
--	---

REVIEWER	Van Steen, Tommy Leiden University
REVIEW RETURNED	20-Dec-2021

GENERAL COMMENTS	The proposed review seems interesting and relevant. As my own research is not in this area, it is difficult to assess how many studies will be found, and as a result, some of the suggestions below might not be feasible due to a low number of studies that can be included in the meta-analytic part of the project. The rationale is clear and the set of databases to be searched is sensible. The overall method seems to work, although I have some suggestions and comments on how to improve the review that I think are required to ensure that all relevant studies (within reason) can be found and included and meaningful conclusions can be drawn. Search related comments:  1. The article summary states that grey literature will not be included. Is there a specific reason for that? I would suggest adding grey literature where possible to paint the complete picture. Furthermore, later on, the authors state that 'ongoing research where datasets are available' will be included, so there seems to be some form of grey literature that is receiving attention anyway. 2. I would add a forward and backward search to the search strategy, checking the reference lists of included articles for any relevant research the search might have missed, and doing the same in a forward sense, where you search the citations of the included papers (e.g. Google Scholar citations) for any newer relevant data. 3. It is useful to specify what the initial search will be carried out on, is it a full text search, or title/abstract only? (Perhaps not every database will support making a specific choice when running the
--

	search.) 4. Is there a rationale for including only adults? (Perhaps because children's health choices might be made by their parents?) I cannot advise on whether or not children should be included, but adding a rationale for the authors' choice would help. Comments relating to other parts of the protocol: 1. The article selection process mentions 4 authors, of which 2 make up the review team with a third being added when the first two disagree. This seems a sensible approach, but as a reader of the protocol I wonder: What does the fourth person do in the article selection process? 2. The authors state that "An effect size greater than 0 indicates that a larger treatment effect was observed in the treatment group relative to the control group." I'm not sure a larger 'treatment' effect can be found in a single group (with between subjects designs it will always be an effect between the two groups), but maybe this is only a wording issue. More importantly however, is that I would suggest to clearly state what a positive effect size means. It would make sense to call a positive effect size an "improvement" in health decisions, but it would help if the authors specified what is considered an improvement. For example, a higher quality of the decision, or a quicker decision. (Mathematically, the latter may result in a negative effect size, where you'd need to correct for that.) 3. The authors need to specify how they weigh the various studies as usually not every study is deemed equally important. The obvious choice would be using the inverse variance, but currently it's unclear which option will be used. 4. Synthesis: I'm assuming you'd first use M/N/SD when you need to calculate the effect size, and only if that's not available use ANOVA/t-test statistics. 5. I'm not sure whether using the mean difference (MD) will be useful in the meta-analysis, as it might be difficult to assess whether domains use the same outcome measures such as decision speed, depending on the decision, this might be minutes/hours/days, and while they use similar outcome measures, it might make more sense to use SMD anyway just in case. Lastly, it might be useful to have a think about which moderators you want to test. Some that come to mind are the risk of bias assessment as this might shine some light on the robustness of the meta-analytical findings, and depending on the type of studies, maybe whether the decision is about the health of the decision maker or whether the decision maker is deciding about someone else's health (e.g. spouse/children). Good luck with the project!
--	--

VERSION 1 – AUTHOR RESPONSE

Reviewer: 1

Dr. Liam Knox, The University of Sheffield

Comments to the Author:

This research proposal is well-written and justified, with the methods exceptionally well documented and appropriate to meet the objectives. The authors should be congratulated on the transparency of this protocol.

I have a total of five suggestions for the authors, four of which only relate to providing greater clarity, and one methodological query. All five are relatively minor.

Rationale

1. Page 3, line 30 and 33: In the description of Wofford, et al., you describe that the main outcome was decisional conflict. However, you do not provide a definition to aid readers. This could be added in parentheses after the first mention. Much in the way you define positivity bias on page 3, line 40.

Author comment: Thank you for highlighting this. We have now included a definition of decisional conflict in the opening paragraph (page 1 lines 51-53) and have included an additional reference for this definition.

2. Page 3, paragraph beginning on line 49: Most of this information appears to be a repeat of what is included in the paragraph beginning line 35. I would suggest that you either combine the two paragraphs or simply remove the second, as you already describe the importance and rationale behind your systematic review.

Author comment: Thank you for spotting this, we have deleted the second paragraph as shown in the track changes on page 2.

3. Hypothesis: your hypothesis is well based upon the literature that you describe; however, you should include the comparator within this. For example, "...audio-visual information would be more beneficial for all patients in comparison to X". If you wanted to go further, you could follow the PICO (participants, intervention, comparison, outcome) format, although this may not be necessary for describing a systematic review hypothesis.

Author comment: We have updated the hypothesis to include the comparator (see page 2 line 45).

Methods and analysis

4. Page 5, line 35: You state that you will extract information on the randomisation process, where this would appear to indicate that you may only be including randomised controlled trials. Whether this is correct or not, a sentence could be added to your 'eligibility criteria' section to provide clarity in this regard.

Author comment: We have updated the 'data items' section and 'eligibility criteria' section to clarify this on page 3 lines 13-14.

5. Effect measures: Is it likely that your included studies will have identical group sample sizes? If not, it would be better to calculate your effect sizes by adjusting your standard deviation with weights for the sample size, or using a correction for the positive bias of Cohen's d (as opposed to simply pooling SD). For the purposes of transparency, it would also be beneficial to explicitly state what effect size you are calculating (Cohen's d, Hedges' g, or Glass' delta).

Author comment: We have now updated the Effects measures and Synthesis methods with the guidance of a statistician. We have stated the effect sizes and correction that we will use (page 5, lines 8-9, and 11-12).

I wish you luck with your review.

Reviewer: 2

Dr. Maria Raisa Aquino, Newcastle University

Comments to the Author:

Dear authors,

Thank you for the opportunity to review this protocol for a systematic review investigating the effectiveness of audio-visual information on health-related decision-making. It is a well-written and designed protocol for a systematic review with meta-analysis. There are a few aspects of this protocol which need clarification and further detail, outlined below. I hope this will be useful to you.

Abstract:br />-No need to list all databases you will be searching in the abstract. Easier to read if you state now many databases you will search with a few examples.

Author comment: Thank you, this has been changed as suggested (Page 1, lines 25-26).

Introduction:

-The rationale for this review is clearly set out.

Methods and analysis:

-You mentioned including ongoing research where datasets are available (page 4, line 16), but have stated in your article summary (page 2, line 52) that you would exclude grey literature. Please can you clarify this?

Author comment: Thank you for highlighting this typo. We have edited the article summary on page 1 which now states that grey literature *will* be included.

-Regarding information sources, you included the Cochrane Database for Systematic Reviews. Will you be including systematic reviews? If so this is not clear and will have implications on your analysis plan.

Author comment: this database was included so that related reviews could be identified and the reference lists of those reviews checked to ensure that other relevant articles are captured. We have added a sentence in 'Article selection process' to explain this decision (page 4, lines 23-24).

You are also planning on including the NHS Economic Evaluation Database, and the relevance of this database to your review aim is unclear. Will you be including a synthesis of the cost-effectiveness of audio-visual information in health-related decision making? If so please include plans for this.

Author comment: Thank you for clarifying, as we will not be including a synthesis of cost-effectiveness we have removed this from our list of databases to be searched.

Finally I wondered if you would do citation searching, or if not, perhaps comment on why you feel this is an unnecessary step.

Author comment: We will do citation searching and have added this as a step on page 4 lines 21-23.

-You might wish to consider what is the added value of searching both MEDLINE and PubMed, or whether the MEDLINE search plus all the other databases (n= 11) you will be searching will be sufficient to capture the extant literature on this topic. The majority of PubMed contain MEDLINE records/citations.

Author comment: Thank you for this suggestion, we would like to include both MEDLINE and PubMed for completeness. Our justification for the inclusion of MEDLINE is that MEDLINE indexes articles using MeSH terms enabling us to get the closest match to our search terms. We would like to include PubMed because as well as containing MEDLINE articles it also contains PubMedCentral papers which includes articles which are not yet in press.

-The search strategy is very brief and could benefit from the input of an information specialist/specialist librarian. Although this is presented as the search strategy for OVID, this will need tweaking for EMBASE as well as MEDLINE. For example: 'exp Decision Making/' returns null results when searching OVID MEDLINE. I think the search strategy presented is also missing a lot of terms related to decision-making, e.g. decision support tools, decision aids, decision support techniques, choice behaviour etc. I strongly recommend that the authors revisit their search strategy in collaboration with an information specialist. This will help to ensure that your review captures as much of the relevant literature.

Author comment: Thank you for your helpful suggestions, we developed the search strategy with the help of a specialist librarian and we have updated the protocol to state this. We used the term 'decision making' as it is a MeSH term and during piloting of search terms we searched the most frequently used keywords in relevant papers. We piloted these search terms in OVID MEDLINE and OVID EMBASE and used the same strategy in both databases. We have updated the protocol to show that these terms can be used in several databases (see table, page 4).

-The article selection process needs further detail. At the moment the full-text screening stage is not described. You have indicated that where there is limited information in the abstract that you would access full-texts. However full-text screening is a separate stage to title and abstract screening. Full-texts of all studies included at title and abstract screening need to be screened to ensure these meet your full eligibility criteria.

Author comment: this has now been clarified on page 4 lines 19-20.

-Data items look good, however I wondered if it would be useful to include what are the decision-making tasks in these studies?

Author comment: Thank you for this important point, decision making task has now been included as a data item on page 4 line 33.

-On the meta-analysis, it would be useful for the authors to comment on how heterogeneity will be addressed.

Author comment: We have now updated the Effects measures and Synthesis methods with the guidance of a statistician and have added in a section on how heterogeneity will be addressed in the 'Synthesis methods' section page 5 lines 23-27.

General feedback on manuscript:

-Full stops in citations with et al. missing, e.g., page 3 line 11: Van der Meulen et al [2].

Author comment: Thank you, this has now been corrected in all instances.

-Page 3, lines 49-53 are a repetition of page 3, lines 35-39. Please check

Author comment: the repeated lines have now been omitted.

Reviewer: 3

Dr. Tommy Van Steen, Leiden University

Comments to the Author:

Comments to the authors

The proposed review seems interesting and relevant. As my own research is not in this area, it is difficult to assess how many studies will be found, and as a result, some of the suggestions below might not be feasible due to a low number of studies that can be included in the meta-analytic part of the project.

The rationale is clear and the set of databases to be searched is sensible. The overall method seems to work, although I have some suggestions and comments on how to improve the review that I think are required to ensure that all relevant studies (within reason) can be found and included and meaningful conclusions can be drawn.

Search related comments:

1. The article summary states that grey literature will not be included. Is there a specific reason for that? I would suggest adding grey literature where possible to paint the complete picture. Furthermore, later on, the authors state that 'ongoing research where datasets are available' will be included, so there seems to be some form of grey literature that is receiving attention anyway.

Author comment: Thank you for highlighting this typo. We have edited the strengths and limitations on page 1 which now states that grey literature *will* be included.

2. I would add a forward and backward search to the search strategy, checking the reference lists of included articles for any relevant research the search might have missed, and doing the same in a forward sense, where you search the citations of the included papers (e.g. Google Scholar citations) for any newer relevant data.

Author comment: thank you for this suggestion, we will do this and have included a sentence stating that this is part of strategy on page 4 under article selection process (page 4 lines 21-23).

3. It is useful to specify what the initial search will be carried out on, is it a full text search, or title/abstract only? (Perhaps not every database will support making a specific choice when running the search.)

Author comment: we have updated the table to show that titles, abstracts and keywords will be searched (see table on page 4).

4. Is there a rationale for including only adults? (Perhaps because children's health choices might be made by their parents?) I cannot advise on whether or not children should be included, but adding a rationale for the authors' choice would help.

Author comment: We only wish to include studies that involve decisions about the self, this is because if one was to make a decision on behalf of someone else there would be additional factors involved and our focus is to understand the influence of audio-visual information on health decision making. We have added a sentence to clarify this on page 3 lines 26-27.

Comments relating to other parts of the protocol:

1. The article selection process mentions 4 authors, of which 2 make up the review team with a third being added when the first two disagree. This seems a sensible approach, but as a reader of the protocol I wonder: What does the fourth person do in the article selection process?

Author comment: We have removed mention of the fourth person as the review team is made up of three people Page 4, line 15).

2. The authors state that "An effect size greater than 0 indicates that a larger treatment effect was observed in the treatment group relative to the control group." I'm not sure a larger 'treatment' effect can be found in a single group (with between subjects designs it will always be an effect between the two groups), but maybe this is only a wording issue.

Author comment: we have changed the wording to reflect that we meant a difference between groups (page 5, lines 12-15).

More importantly however, is that I would suggest to clearly state what a positive effect size means. It would make sense to call a positive effect size an "improvement" in health decisions, but it would help if the authors specified what is considered an improvement. For example, a higher quality of the decision, or a quicker decision. (Mathematically, the latter may result in a negative effect size, where you'd need to correct for that.)

Author comment: we have clarified what an improvement in decision making would mean on page 5, lines 12-15.

3. The authors need to specify how they weigh the various studies as usually not every study is deemed equally important. The obvious choice would be using the inverse variance, but currently it's unclear which option will be used.

Author comment. We have now added a line stating we will use inverse variance on page 5 line 21.

4. Synthesis: I'm assuming you'd first use M/N/SD when you need to calculate the effect size, and only if that's not available use ANOVA/t-test statistics.

Author comment: yes we have changed this on page 5 lines 18-19.

5. I'm not sure whether using the mean difference (MD) will be useful in the meta-analysis, as it might be difficult to assess whether domains use the same outcome measures such as decision speed, depending on the decision, this might be minutes/hours/days, and while they use similar outcome measures, it might make more sense to use SMD anyway just in case.

Author comment: We have omitted speed of decision making as a measure of interest as on reflection we decided it detracted from the focus of the review. We agree that it's likely that studies will use different measures and have changed the analysis section throughout to state that SMDs will be used.

Lastly, it might be useful to have a think about which moderators you want to test. Some that come to mind are the risk of bias assessment as this might shine some light on the robustness of the meta-analytical findings, and depending on the type of studies, maybe whether the decision is about the health of the decision maker or whether the decision maker is deciding about someone else's health (e.g. spouse/children).

Author comment. Thank you for this suggestion, we will include risk of bias as a moderator (page 5, lines 26-27).

Good luck with the project!

VERSION 2 – REVIEW

REVIEWER	Knox, Liam The University of Sheffield, Department of Neuroscience
REVIEW RETURNED	10-Mar-2022
GENERAL COMMENTS	Thank you for your response to each of our reviewer comments. I believe the protocol is ready for publication. Good luck with your research.